

# High abundance of Early Miocene sea cows from Qatar shows repeated evolution of seagrass ecosystem engineers in Eastern Tethys

Nicholas D. Pyenson[1], Ferhan Sakal[2], Jacques LeBlanc[3], Jon Blundell[4], Katherine D. Klim[5], Christopher D. Marshall[6,7], Jorge Velez-Juarbe[8], Katherine Wolfe[4] and Faisal Al-Naimi[2]

[1] Department of Paleobiology, National Museum of Natural History, Smithsonian Institution, Washington, D.C., United States of America
[2] Department of Archaeology, Qatar Museums, Doha, Qatar
[3] Panama City, Panama
[4] Digitization Program Office, Office of Digital Transformation, Smithsonian Institution, Washington, D.C., United States of America
[5] Stone Ridge School of the Sacred Heart, Bethesda, MD, United States of America
[6] Department of Marine Biology, Texas A&M University at Galveston, Galveston, TX, United States of America
[7] Department of Ecology and Conservation Biology, Texas A&M University, College Station, TX, United States of America
[8] Department of Mammalogy, Natural History Museum of Los Angeles County, Los Angeles, CA, United States of America

Corresponding author
Nicholas D. Pyenson,
pyensonn@si.edu

## ABSTRACT

Coastal ecosystems that include seagrasses are potential carbon sinks that require strategic conservation of top trophic consumers, such as dugongs, to maintain their function. It is unclear, however, how long seagrass ecosystems have persisted in geologic time because their fossil record is poor, although the record of their associated vertebrate consumers offers useful proxies. Here we describe an area of dense Early Miocene dugongid remains from Qatar. We documented over 172 sites in <1 km$^2$ from one stratigraphic level, including material representing a new species of fossil dugongine dugongid. This taxon is unrelated to coeval Early Miocene dugongids from India and the Eastern Tethys and it is distantly related to extant dugongs, which occupy seagrass habitats of the Persian or Arabian Gulf (hereafter 'Gulf') today. The monodominant assemblage in this area likely reflects a single fossil dugongid taxon and matches the ecological diversity and geospatial distribution of modern-day live-dead assemblages in the Gulf. This fossil site from Qatar shows that the Gulf has repeatedly evolved sea cow communities with different taxa over the past 20 million years and coincides with an Early Miocene marine biodiversity hotspot in Arabia, prior to its eastward shift to the Indo-Australian Archipelago where dugongs continue to thrive today.

# INTRODUCTION

Marine mammals play key roles in maintaining ocean health through their abundance as high-trophic level consumers or as keystone species (*Hazen et al., 2024*). In coastal ecosystems, seagrasses are major carbon sinks that have strong potential as natural climate solutions (*Fourqurean et al., 2012*). Notably, the species richness and abundance of seagrasses are maintained by marine vertebrates that forage in these communities, such as sea turtles (*i.e., Chelonia mydas*) and dugongs (*i.e., Dugong dugon*). These marine herbivores function as ecosystem engineers to create habitat while foraging on seagrasses. In particular, dugongs excavate feeding trails and pits that enhance nutrient availability and cycling (*Coleman & Williams, 2002*) through bioturbation generated by their feeding biomechanics (*Marshall et al., 2003*).

Today, dugongs range from Oceania through southern Asia to coastal Africa, but one of the largest aggregations occurs in the Gulf, along coastal habitats from Saudi Arabia, Bahrain, Qatar, and the United Arab Emirates (*Marshall et al., 2018*; *Khamis et al., 2023*). While many of the details about the migration and size of this transboundary population remain unclear, Gulf dugongs number in the hundreds of individuals. Dugongs in the Gulf are also directly threatened by regional fisheries bycatch, coastal development, and desalination projects (*Kawiyani et al., 2024*), which along with climate change, pushes many organisms near their physiological limits during the summer in the Gulf (*Kawiyani et al., 2024*; *Marshall et al., 2020*). Forecasting the biological response of seagrass communities to future climate projections, especially in the Gulf (*Al-Mudaffar Fawzi et al., 2022*; *Fieseler et al., 2023*) will require comparable datasets from the geologic past. However, it is unclear how long marine herbivores have been ecological engineers in coastal communities (*Steneck, Bellwood & Hay, 2017*), especially given the terrestrial ancestries of sirenians in the early Cenozoic and sea turtles in the early-mid Mesozoic (*Kelley & Pyenson, 2015*). For sirenians in particular, their excellent fossil record shows an Eocene or Oligocene antiquity to this ecological role (*Domning, 2018*), but data supporting this argument are sparse.

Fossil seagrasses are rare and have low preservation potential (*Beavington-Penney, Wright & Woelkerling, 2004*), yet it has been proposed that the presence of these communities in the geologic past can be inferred by the fossil record of associated marine herbivores (*Vélez-Juarbe, 2014*). Fossil sea turtles with seagrass-shearing ecomorphologies have evolved repeatedly since the Cretaceous (*Parham & Pyenson, 2010*), indicating a large potential geologic interval for seagrass foraging; sea cows evolved from semiaquatic terrestrial ancestors in the Eocene, and became obligately aquatic by the late Eocene (*Domning, 2018*). How sirenians evolved underwater foraging to become ecosystem engineers remains unclear. Here we report a monodominant bonebed assemblage of fossil dugongids from the Early Miocene Dam Formation of Qatar. The abundance and geospatial distribution of these fossil dugongids, along with the phylogenetic analysis of the presumed monodominant taxon, suggest that this assemblage represents the first of several marine herbivore invasions of coastal ecosystems along the Arabian Peninsula in the past ~20 million years, coinciding with a marine biodiversity hotspot that has migrated across Europe and Asia during the Cenozoic (*Renema et al., 2008*).

## MATERIALS & METHODS

### Fieldwork

From 2023–2024, we prospected fossil-bearing outcrops of the Dam Formation throughout southwestern Qatar under permits from Qatar Museums and the Ministry of Environment and Climate Change in Qatar. With the support of these institutions, we accessed the area of Al Maszhabiya and other fossil localities in the Southern Reserve, which is located behind secure fencing in an area east and south of the Salwa Road in southwestern Qatar, north of the border with Saudi Arabia. The area of Al Maszhabiya is adjacent to the Al Eraiq Reserve and the eastern border of the vehicle exclusion zone (Fig. 1) overlaps slightly with the Al Eraiq Reserve. Both the Al Maszhabiya area and the Al Eraiq Reserve are located within the Southern Reserve. The Al Maszhabiya area is registered as Heritage Area 23400 for Qatar Museums and protected by the Ministry of Environment and Climate Change. Reports of Work describing our field seasons are archived with Qatar Museums.

### Sedimentology and geologic age

The fossil dugongid-rich outcrops of the Al Maszhabiya bonebed are restricted to a horizon in the Lower Al-Kharrara Member, which is one of two members that belong to the lower part of the Early Miocene Dam Formation in southwest Qatar (*Cavelier, 1970*; *LeBlanc, 2021*). The Lower Al-Kharrara Member is part of a silicate-dolomite-calcite sequence in the Dam Formation in southwest Qatar (*Al-Saad & Ibrahim, 2002*). In the Al Maszhabiya area, this member exposes fine-grained siliciclastics (levels 17–20 of *Dill & Henjes-Kunst, 2007*), representing a shallow marine environment preserving abundant fossil molluscs. The bonebed is overlain by a calcitic clay-rich marlstone (20–22 of *Dill & Henjes-Kunst, 2007*), which represents an inter-tidal to beach environment. Within the Lower Al-Kharrara Member, horizontal stratification with even bedding planes and bedsets measuring up to 1 m is widespread, particularly in the siltstones and fine-grained sandstones, which are overlain by a calcitic clay-rich marlstone. We constrain the age of the Al Maszhabiya bonebed to 23.03–21.6 Ma based on $^{87}Sr/^{86}Sr$ ratios of calcareous and evaporitic marine sediments in the Dam Formation (*Dill & Henjes-Kunst, 2007*; see Supplemental Information text for more details).

### Phylogenetic analysis

We used the character-state matrix from *Suárez et al. (2021)*, which was derived from *Vélez-Juarbe & Wood (2018)*. For more details on outgroup and ingroup selection, along with specific specimens used for taxa coded in our character-state matrix, see Supplemental Information text. We performed the analysis in TNT v.1.5 (*Goloboff, Farris & Nixon, 2008*); all characters were initially treated as unordered and analyzed under equal and implied weights ($k = 3$ and $k = 9$; see *Goloboff et al., 2008* and *Goloboff, Torres & Arias, 2018*). We then performed a heuristic search of 10,000 replicates using the tree bisection-reconnection (TBR) algorithm with a backbone constraint tree based on the molecular phylogeny from *Springer et al. (2015)*. Bootstrap values were obtained by performing 10,000 replicates.
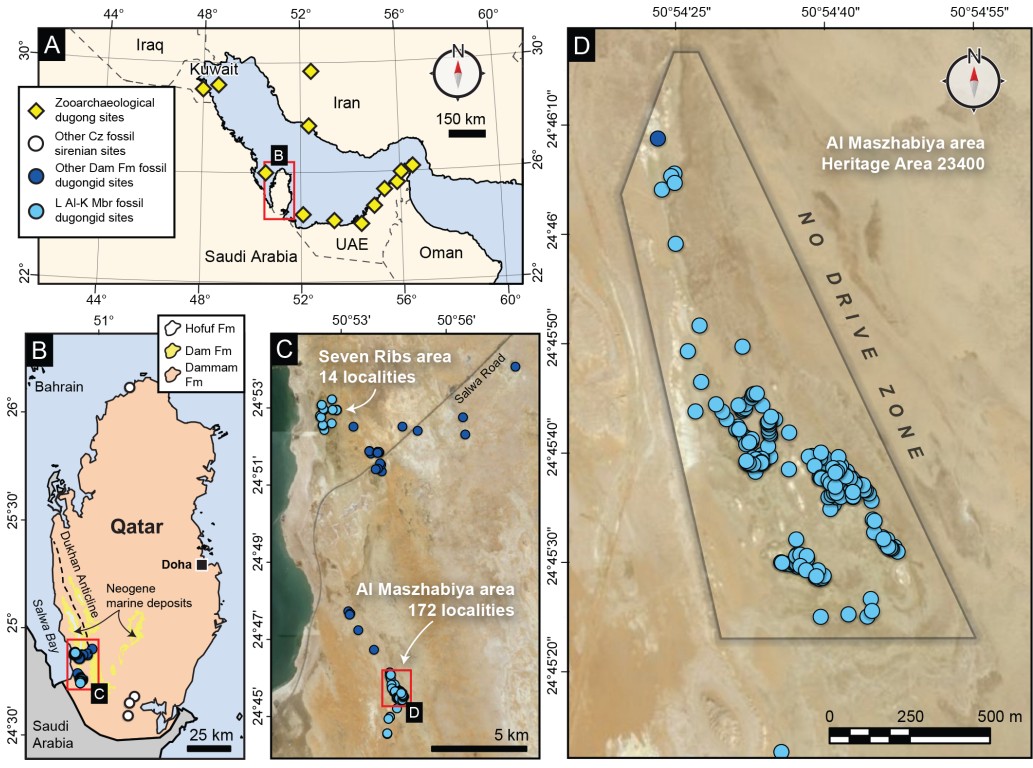

**Figure 1** **Geographic context of fossil dugongids from Qatar.** (A) The Gulf; (B) Qatar with fossil dugongid localities; (C) fossil dugongid localities in southwest Qatar; (D) fossil dugongid localities at Al Maszhabiya. Open white dots denote non-Miocene Cenozoic localities, dark blue dots are Dam Formation fossil dugongid localities except for light blue localities belonging to dugongids from the Lower Al-Kharrara Member. See Supplemental Information text for more details.

## Nomenclatural acts

The electronic version of this article in Portable Document Format (PDF) will represent a published work according to the International Commission on Zoological Nomenclature (ICZN), and hence the new names contained in the electronic version are effectively published under that Code from the electronic edition alone. This published work and the nomenclatural acts it contains have been registered in ZooBank, the online registration system for the ICZN. The ZooBank LSIDs (Life Science Identifiers) can be resolved and the associated information viewed through any standard web browser by appending the LSID to the prefix http://zoobank.org/. The LSID for this publication is: urn:lsid:zoobank.org:pub:14F64E75-9D4C-4F3D-B0A6-62D4D5863B13. The online version of this work is archived and available from the following digital repositories: PeerJ, PubMed Central and CLOCKSS.

## Aerial and orthographic 3D datasets from field localities

We documented *in situ* skeletal remains of dugongids from Al Maszhabiya bonebed using aerial photography and three-dimensional digitization techniques (*Pyenson et al., 2014*). With permission from the Ministry of Environment and Climate Change in Qatar, we

collected aerial photography in 2024 with a DJI MAVIC 3 Pro, including video and still images from the area (Fig. S4). Photogrammetry datasets from both field and museum settings were captured using a prime 35 mm Canon L series lens on a Canon 5D Mark III camera body. Scale was set for the photogrammetry data using scale bars designed by the United States Bureau of Land Management and produced and calibrated by Cultural Heritage Imaging. We used X-Rite ColorChecker targets for color calibration of the photogrammetry image sets and produced color corrected images using the X-Rite ColorChecker software and Adobe Camera Raw. Agisoft PhotoScan 2.0 was used for photogrammetry model creation; Geomagic Studio 2021 for model cleanup and noise reduction.

### High-resolution μCT scanning

To resolve inner morphology of ARC.2023.28.014, we scanned it using the GE Phoenix v|tome|x M 240/180 kV Dual Tube 3D computed tomography at the Micro Computed Tomography Imaging Center (mCTIC) at the Smithsonian Institution's National Museum of Natural History (NMNH) in Washington, D.C., USA. Voxel sizes ranged from 2–4.5 μm. The raw CT data were reconstructed using GE Datos 2 and slices were exported using VG Studio Max 3.2. Final images were then resolved using Adobe Photoshop 2022.

### Data, materials, and software availability

Source datasets for paleoecological and taphonomic analyses, digital image database of the FD 23 localities, orthographic images of field localities, and phylogenetic analyses have been deposited on Zenodo (DOI: 10.5281/zenodo.15312915).

Source datasets for 3D shapefiles and raw CT data are deposited at Morphosource (Project ID: 000747006) and are accessible at: https://www.morphosource.org/projects/000747006?locale=en.

## RESULTS

### Locality and geological context

We documented 172 localities bearing dugongid skeletal material in the Al Maszhabiya area (Fig. 1). We also identified fossil cetaceans (Odontoceti indet.), fossil turtle (Testudines indet.), and fossil teleosts and carcharhiniform sharks at some of these localities (Fig. 2; Fig. S1). All of these localities belong to an area of 0.76 km$^2$ bounded by a vehicle exclusion zone near the coordinates of 24°45'36.3''N, 50°54'36.2''E in Al Rayyan Municipality of the State of Qatar. The area of Al Maszhabiya is adjacent to the Al Eraiq Reserve and the eastern border of the vehicle exclusion zone overlaps slightly with the Al Eraiq Reserve. Both the Al Maszhabiya area and the Al Eraiq Reserve are located within the protected Southern Reserve. The Al Maszhabiya area is registered as Heritage Area 23400 for Qatar Museums and protected by the Ministry of Environment and Climate Change in Qatar. The fossil-bearing localities at Al Maszhabiya belong to a single package of fine-grained siliciclastic sediments about 0.5 m thick belonging to the Lower Al-Kharrara Member of the Dam Formation (*sensu LeBlanc, 2021*; see *Cavelier, 1970*; *Al-Saad & Ibrahim, 2002*; *Dill & Henjes-Kunst, 2007*; Fig. 2).

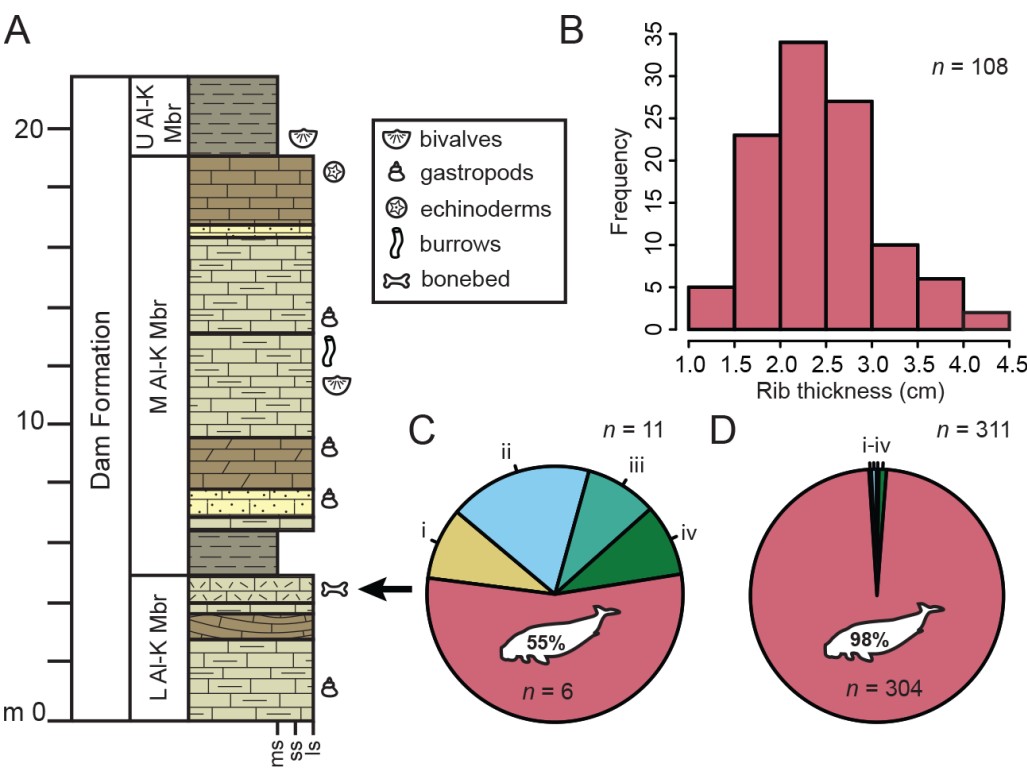

**Figure 2 Stratigraphic and taphonomic context of Al Maszhabiya bonebed.** (A) Section of the Lower Al-Kharrara Member (L Al-K Mbr) of the Dam Formation and summary stratigraphy based on *Cavelier (1970)* and *LeBlanc (2021)* with the stratigraphic position of the Al Maszhabiya bonebed; M Al-K Mbr and U Al-K Mbr, Middle and Upper Al-Kharrara members, respectively; ms, mudstone; ss, sandstone; ls, limestone. Arrow denotes the bonebed. (B) Distribution of maximum rib thickness for 108 fossil dugongids. (C) Minimum number of individuals and (D) minimum number of elements based on fossil vertebrates, most of which are fossil dugongids (percentage and total near silhouette). Other vertebrates are (i) Odontoceti; (ii) Testudines; (iii) Carcharhiniformes; and (iv) Osteichthyes. See Supplemental Information text for more details.

## Systematic Paleontology

Mammalia *Linnaeus, 1758*
Afrotheria *Stanhope et al., 1998*
Tethytheria *McKenna, 1975*
Sirenia *Illiger, 1811* sensu *Vélez-Juarbe & Wood, 2018*
Dugongidae *Gray, 1821* sensu *Vélez-Juarbe & Wood, 2018*
Dugonginae *Gray, 1821* (*Simpson, 1932*) sensu *Vélez-Juarbe & Wood, 2018*

*Salwasiren qatarensis* gen. et sp. nov.
urn:lsid:zoobank.org:act:865814BD-0B78-403F-B64F-4F77363762A6

## Etymology

"Salwa" after the Bay of Salwa, part of the transboundary habitat for dugongs in the Gulf combined with the Latin "siren", referring to Sirenia. The species epithet honors its discovery in the State of Qatar.

## Holotype

Qatar Museums ARC.2023.23.008, representing one associated skeleton including an incomplete cranium, mandible, left upper molar (M2), sternum, both scapulae and humeri, a partial vertebral column and ilium from locality FD 23-56 (Fig. 3, Fig. S2).

Incomplete cranium 3D shapefiles at Morphosource:https://doi.org/10.17602/M2/M747768
Incomplete mandible 3D shapefiles at Morphosource: https://doi.org/10.17602/M2/M773045
Sternum 3D shapefiles at Morphosource: https://doi.org/10.17602/M2/M773049
Left upper molar (M2) 3D shapefiles at Morphosource: https://doi.org/10.17602/M2/M773058
Right scapula 3D shapefiles at Morphosource: https://doi.org/10.17602/M2/M773062
Left scapula 3D shapefiles at Morphosource: https://doi.org/10.17602/M2/M773066
Right humerus 3D shapefiles at Morphosource: https://doi.org/10.17602/M2/M773074
Left humerus 3D shapefiles at Morphosource: https://doi.org/10.17602/M2/M773070
Axis, second cervical vertebra (C2) 3D shapefiles at Morphosource: https://doi.org/10.17602/M2/M773078
Thoracic vertebra (Ta) 3D shapefiles at Morphosource: https://doi.org/10.17602/M2/M773082
Lumbar vertebra (La) 3D shapefiles at Morphosource: https://doi.org/10.17602/M2/M773086
Lumbar vertebra (Lb) 3D shapefiles at Morphosource: https://doi.org/10.17602/M2/M773090
Lumbar vertebra (Lc) 3D shapefiles at Morphosource: https://doi.org/10.17602/M2/M773094
Sacral vertebra 3D shapefiles at Morphosource: https://doi.org/10.17602/M2/M773098.
Caudal vertebra (Ca) 3D shapefiles at Morphosource: https://doi.org/10.17602/M2/M773102
Caudal vertebra (Cb) 3D shapefiles at Morphosource: https://doi.org/10.17602/M2/M773106
Caudal vertebra (Cc) 3D shapefiles at Morphosource: https://doi.org/10.17602/M2/M773110
Caudal vertebra (Cd) 3D shapefiles at Morphosource: https://doi.org/10.17602/M2/M773115
Caudal vertebra (Ce) 3D shapefiles at Morphosource: https://doi.org/10.17602/M2/M773119
Caudal vertebra (Cf) 3D shapefiles at Morphosource: https://doi.org/10.17602/M2/M773135
Left ilium 3D shapefiles at Morphosource: https://doi.org/10.17602/M2/M773140

## Referred material

ARC.2023.28.014, an incomplete left incisor (I1).

Incomplete left incisor 3D shapefiles at Morphosource: https://doi.org/10.17602/M2/M747752
Incomplete left incisor Raw CT data at Morphosource: https://doi.org/10.17602/M2/M747692

### Type locality, horizon, and age

Al Maszhabiya bonebed, Lower Al-Kharrara Member of the Dam Formation, Aquitanian, 23.03–21.6 Ma.

### Differential diagnosis

*Salwasiren* is a dugongine distinguished from other sirenians by the following combination of characters: nasal process of the premaxilla long, thin and tapering at posterior end (c.6[0], 7[0]) as in *Crenatosiren olseni* and *Dugong dugon*; supraorbital process of frontal dorsoventrally thick with a weakly developed posterolateral corner (c.36[1]), as in *C. olseni* and *D. dugon*; deep and narrow nasal incisure (c.37[1]) as in most dugongines; flat frontal roof (c.42[0], as in *C. olseni*, *Italosiren bellunensis* and *Bharatisiren indica*; supraoccipital wider ventrally than dorsally (c.23[1]) and exoccipitals not meeting along a dorsal suture (c.66[1]), as in *Nanosiren* spp. and *D. dugon*; ventral extremity of jugal under posterior edge of orbit (c.85[1]) and flat, thin preorbital process of jugal (c.88[0]), shared with *C. olseni* and *Nanosiren* spp.; short zygomatic process of the jugal (c.89[1]), as in *Dioplotherium manigaulti* and *Xenosiren yucateca*; ventral rim of orbit that does not overhang the lateral surface (c.90[0]), as in *I. bellunensis* and *Callistosiren boriquensis*; mandible with broad, subrectangular symphysis (c.121[3]); I1 alveolus small (c.140[0]) as in *Nanosiren* spp.; I1 with suboval cross section and enamel on all sides (c.141[0], 142[0]), as in *C. olseni* and *N. sanchezi*; pubis prong-like without symphysis (c.215[2]).

### Description

The type specimen of *Salwasiren qatarensis* (ARC.2023.23.008) consists of one associated skeleton including an incomplete cranium, mandible, upper molar (M2), sternum, both scapulae and humeri, a partial vertebral column and ilium (Fig. 3).

The cranium is incomplete and includes a partial left premaxilla that is missing a portion of its distal termination, but medially it does not show an alveolus for I1; the putative size of I1 in *Salwasiren* is therefore small (c.139[0], 140[0]) and consistent with the tusk morphology we identified in the referred left I1 (ARC.2023.28.014), which was also collected from the Al Maszhabiya bonebed. The premaxilla also preserves a notable boss (c.10[1]) demarcating the deflection of the distal rostrum similar to *Callistosiren* and most other dugongids. The premaxilla in *Salwasiren* tapers posteriorly and it is thin, not broad nor bulbous (c.6[0]). The premaxilla does not form a butt joint with the frontal as it does in *Dioplotherium*. The nasal opening is long and retracted (c.8[1]); the nasals in *Salwasiren* have fused with the frontals during ontogeny but they are separated at the midline (c.31[1], 32[2]), as in most other dugongines. The nasal incisure is deeper than the level of the postorbital process of the frontal (c.37[1]), which has a weakly developed posterolateral corner (c.36[1]). The frontal roof lacks bosses and it is convex between the posterior termination of the nasal incisure and the contact with the parietals (c.42[0], 45[0]). The parietals are nearly flat (c.51[1]), with relatively low temporal crests (type B of *Domning, 1988*). Internally the parietals show a prominent falx cerebri that extends posteriorly to reach the internal occipital protuberance (c.223[0], 225[1]); the tentorium osseum extends laterally from the internal occipital protuberance (c.224[1]).

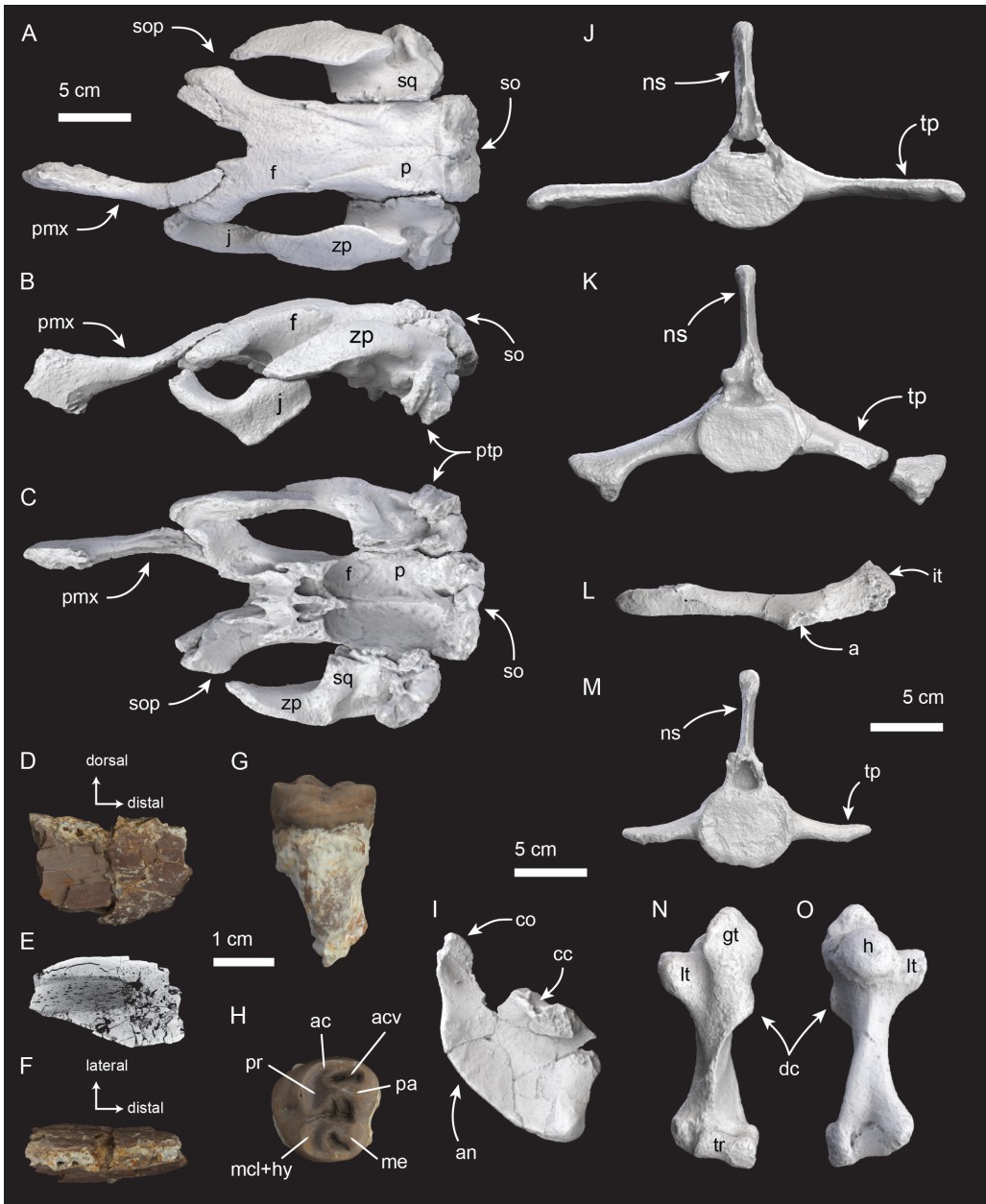

**Figure 3** *Salwasiren* **morphology using 3D photogrammetry of μCT of key skeletal elements.**
(A–C) holotype cranium including left premaxilla, jugal, and partial braincase; (D–F) referred left incisor; (G–H) holotype left upper M2, in mesial and occlusal views, respectively; (I) holotype incomplete left mandible; (J–M) holotype lumbar, sacral, and caudal vertebrae in anterior views with left ilium (L) in lateral view; and (N–O) holotype right humerus. Abbreviations: a, acetabulum; ac, anterior cingulum; acv, anterior cingular valley; an, angular process; co, coronoid process; cc, coronoid canal; dc, deltoid crest; f, frontal; gt, greater tubercle; h, humeral head; it, ischial tuberosity; j, jugal; lt, lesser tubercle; mcl+hy, metaconule + hypocone; me, metacone; ns, neural spine; p, parietal; pa, paracone; pmx, premaxilla; pr, protocone; ptp, posttympanic process; so, supraoccipital; sop, supraorbital process; sq, squamosal; tp, transverse process; tr, trochlea; zp, zygomatic process.

In posterior view, the supraoccipital is wide at its base, like *Nanosiren* and *Dugong*, but the exoccipitals are separated and do not meet at the midline (c.64[1], 66[1]). In lateral view, the zygomatic process of the squamosal gradually tapers anteriorly (c.81[0]), and its posterior end exhibits a slight ventral deflection (c.77[1]), whereas this trait is horizontal in *Hydrodamalis* and more ventrally deflected in *Nanosiren*. In dorsal view, the zygomatic processes are medially concave as in most dugongids (c.84[0]). On the squamosal, the external auditory meatus exhibits a semicircular opening (oriented anteroventrally) in lateral view (c.82[1]), the sigmoid ridge is present and prominent (c.74[1]). Lastly, the cranial portion of the squamosal, at its articulation with the cranium, extends dorsally to the temporal crest (c.76[1]). The preorbital process of the jugal is overall relatively flat (c.88[0]). Anteriorly, the jugal has a notch for the lacrimal on its preorbital process; the jugal also shows that the orbit is longer than the zygomatic process of the jugal (c.89[1]). The ventralmost point of the jugal is posterior of the level of the orbit, as in most Paleogene sirenians, in contrast to it being aligned in most Miocene dugongines (c.85[1]). On the lateral surface of the jugal, the ventral rim of the orbit is bulbous, but it does not markedly overhang the lateral surface as it does in other dugongines (c.90[0]). While no lacrimal was identified among the skeletal remains, the notch on the jugal suggests that it was large, and we have coded this trait using this feature (c.91[?1]). No maxilla is represented in the type specimen of *Salwasiren*; there is a possible pterygoid, which will be described in more detail with future work. While there was no alisphenoid preserved with the material, the tympanic was clearly unfused, as it is sitting in a close-fitting socket within the squamosal (c.115[1]) as in most other sirenians.

The type is also represented by an incomplete left mandible preserving the mandibular condyle, the coronoid canal, and the mandibular canal. The ventral angle is not preserved, but the mandibular capsule is exposed posteroventrally (c.127[1]). The mandibular condyle faces dorsally and the posterior border is broadly convex (c.125[2]). We refer an isolated mandible (ARC.2024.28.021) collected at locality FD 23-147 in 2024 to *Salwasiren* based on similar morphology. This mandible shows a fused symphysis typical of dugongids. For dentition, the upper left molar (M2) associated with the type specimen is roughly as wide as it is long in occlusal view and shows smooth cheek tooth enamel (c.151[0], 156[0]). The referred left I1 (ARC.2023.28.014) is incomplete but it is sub-elliptical to square in cross-section, showing enamel its entire length and on all sides (c.141[0], 142[0]). Based on the premaxilla with the type specimen, the upper incisor did not extend into the length of the premaxilla in *Salwasiren*, which is consistent with the small size of the referred incisor.

For the postcranium, the type of *Salwasiren* includes both scapulae and humeri. The humeral head has a bicipital groove and the rest of the humerus is slender, not comparatively bulky, and has a prominent, recurved deltoid process (c.213[0], 221[0], 222[1]). Based on the associated pelvis, the pubis in *Salwasiren* is unfused and prong-like with a clear acetabulum that suggests a reduced femur (c.215[2], 217[1]). The vertebral series in *Salwasiren* is represented by a second cervical (axis, C2), an incomplete thoracic, three lumbar, one sacral, and five caudal vertebrae with a single fused and single unfused chevron (c.204[1], 205[1]). It is unclear if the three lumbar and five caudal vertebrae from the type

specimen belong in adjacent sequences. The spinous processes on the lumbars do not have flanges like *Pezosiren* (c.203[0]).

## Paleoecology

The exposed strata in the Al Maszhabiya area includes a monodominant bonebed, represented overwhelmingly by fossil dugongids, distributed in a 0.5 m-thick limestone layer with abundant oysters and bivalves and associated fossil vertebrates at the top of the Lower Al-Kharrara Member of the Dam Formation. At each fossil dugongid locality, we collected calibrated digital images of the exposed skeletal elements for taphonomic analyses. The Al Maszhabiya bonebed is mostly represented by dugongid ribs (84% of all diagnostic skeletal remains, see Table S1), with a few associated or semi-articulated individuals. Taphonomic scoring of skeletal articulation (*Pyenson et al., 2014*) for 179 fossil vertebrates revealed the majority (77%) to be isolated or separate elements, with only 24% represented by associated but disarticulated elements, and none represented by articulated elements (see Table S2). We found that most elements were lightly abraded, with occasional bioencrustation; we did not identify any bite marks or similar traces (see Table S3).

Based on the abundance of ribs (see Figs. S3–S4), we measured maximum rib thickness from available ($n = 108$) localities, revealing an essentially normal size distribution of ribs 1.0–4.5 cm thick (Fig. 2), with highest frequency occurring between 2.0–2.5 cm (roughly the size of modern *Dugong*; see Tables S4–S5, Fig. S5). Based on an articulated digital model of the *Salwasiren* type cranium (Fig. 3), we calculated body length of 200 cm, using the dugong equation of *Sarko et al. (2010)*. This estimated body length for *Salwasiren* is roughly comparable to a sub-adult *Dugong*, but larger than *Nanosiren* estimates. No rib material was associated with the type of *Salwasiren*.

Based on our current survey of 172 fossil dugongid localities, we calculated a MNI (minimum number of individuals) of 6 individuals to account for 304 dugongid skeletal elements identified at Al Maszhabiya; dugongid MNI accounts for 54% of the documented fossil vertebrate abundance. By contrast, the 304 MNE (minimum number of elements) accounts for 98% of the fossil vertebrate skeletal remains (see Table S6). We think that the high abundance of dugongids at Al Maszhabiya is best described by MNE because the close association of disarticulated skeletal remains from three localities (*i.e.,* FD 23-14, the type *Salwasiren* locality FD 23-56, and FD 23-75; Fig. S6) is closest to the skeletal distribution of dugong carcasses observed for beach carcasses in the area of northwest Qatar and the Hawar Islands of Bahrain (see Fig. S2, Tables S7–S9). This modern comparison, along with the high number (>30 localities) with multiple ribs in association, supports the argument that dugongid fossil sites >5 m apart at Al Maszhabiya likely represent individual skeletons. Thus, we argue that the abundance of individual fossil dugongids from Al Maszhabiya is likely closer to MNI $\approx 10^2$. Lastly, the MNI and MNE values at Al Maszhabiya exceed those in a 1.1 km$^2$ area of the Seven Ribs localities (MNI = 1, MNE = 14) about 25 km northeast in the same stratigraphic horizon of the Lower Al-Kharrara Member of the Dam Formation (see Table S10).

## Phylogenetic results

Using a modified version of the matrix of fossil and living sirenians from *Suárez et al. (2021)* with a molecular backbone (see Supplemental Information text), we recovered identical topologies using implied weights of $k = 9$ (180 MPTs of fit = 14.41) and $k = 12$ (180 MPTs of fit = 11.65). The consensus tree places *Salwasiren* in a clade of fossil dugongine dugongids related to extant *Dugong* and sister to a dugongine clade of exclusively fossil taxa, including a polyphyletic assemblage of Indian taxa from the Early Miocene Khari Nadi Formation (Fig. 4). Our analysis was largely consistent with other recent ones (*i.e., Suárez et al., 2021*; *Díaz-Berenguer et al., 2018*; but see *Heritage & Seiffert, 2022*). We recovered a consistent distribution of stem sirenians (including Protosirenidae), outside crown Sirenia (*Trichechus* + *Dugong*) that includes crown Dugongidae (*Hydrodamalis* + *Dugong*) as defined by *Vélez-Juarbe & Wood* (*2018*; see also Fig. S4). In our analyses, the position of *Salwasiren* was well supported by five synapomorphies, including a ventrally broad supraoccipital, a short zygomatic process of the jugal and a prong-like pubis. Its position was sensitive to the coding of the referred incisor (ARC 2023.28.014) which was collected from the bonebed at a separate locality (FD 23-121) about 230 m southwest of the holotype. The resulting topology did not change when adding the geographic distribution character/states from *Heritage & Seiffert (2022)*.

## DISCUSSION

### Paleoecology

The Aquitanian fossil vertebrate assemblage from the Al Maszhabiya area represents a monodominant bonebed consisting primarily of isolated postcranial dugongid remains (*i.e.,* ribs and other postcranial elements). Dugongid fossils dominate the assemblage, which includes abundant reef-building marine invertebrate fossils (*i.e.,* oysters). These fossils represent a shallow marine environment with water depths between 5–25 m. Although we documented no fossil seagrasses from the Lower Al-Kharrara Member, this paleoenvironmental interpretation is analogous with the nearshore depositional settings today directly adjacent to Saudi Arabia and Qatar in the Bay of Salwa and seagrass meadows between Bahrain and the Hawar Islands. The latter marine environment off Qatar's northwestern coastline today sustains abundant marine herbivores, such as dugongs and green sea turtles, which dominate the adjacent coastal stranding record of Qatar.

We propose that the Al Maszhabiya bonebed represents a time-averaged attritional assemblage, formed over the course of $10^4$ years, given the sedimentation rate for the Dam Formation (3 to 4 cm/kyr; see *Alkhaldi, Read & Al-Tawil, 2021*). Multiple lines of evidence imply intermittent exposure on a seafloor or a dispersal of elements during or prior to burial in low-energy setting, including: the absence of intact, articulated axial or appendicular dugongid remains in the bonebed; the slight abrasion on 94–95% of the fossil material; and evidence of oyster encrustation on some dugongid material (*El-Hedeny, 2007*; see Table S3, Fig. S2). The proximity of the completely disarticulated elements of the type specimen of *Salwasiren* is directly comparable to modern dugong carcasses observed in littoral zones of northwest Qatar and the Hawar Islands of Bahrain (see Fig. S2,

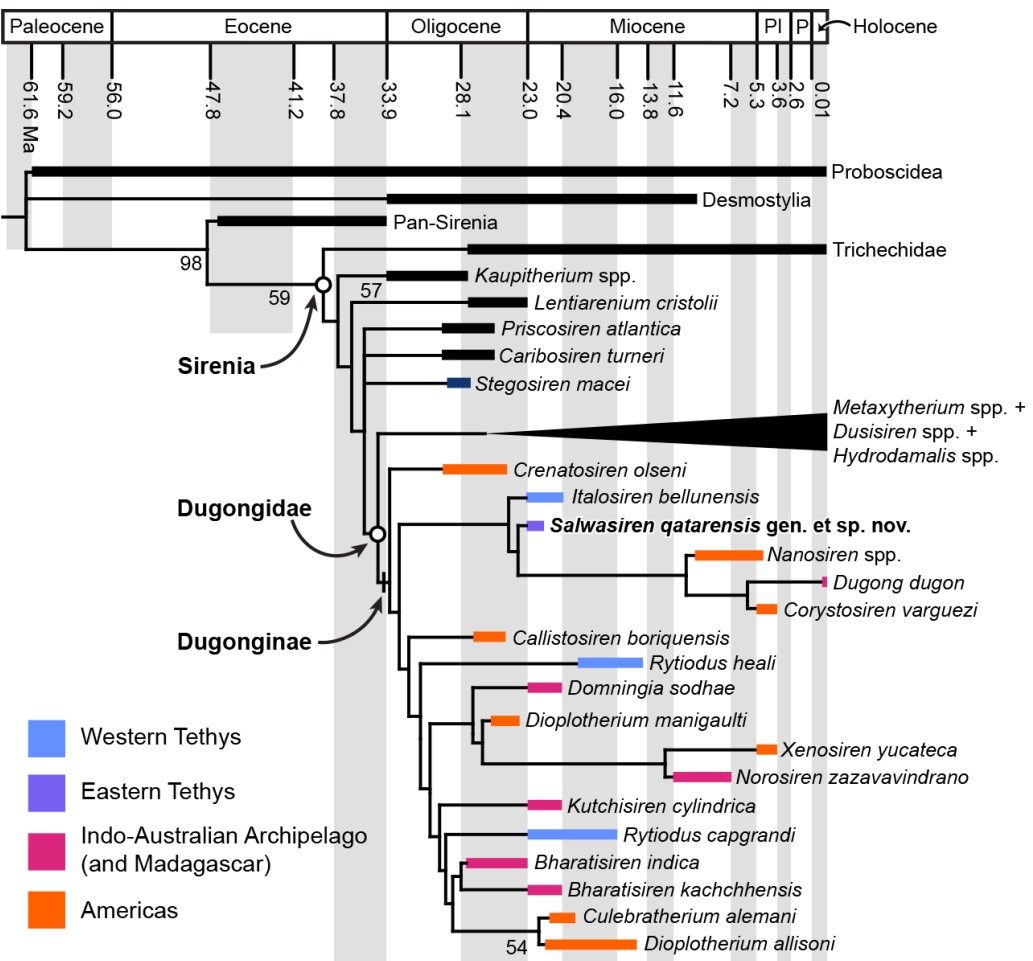

**Figure 4** **Time-calibrated strict consensus tree of Sirenia (180 MPTs of fit = 14.41).** Pan-Sirenia, Trichechidae, and hydrodamalines are collapsed for ease of comparison; see full tree in Fig. S7. Ages and clade names follow *Suárez et al. (2021)* and *Vélez-Juarbe & Wood (2018)*, respectively. Numbers at nodes indicate bootstrap values. Abbreviations: Pl, Pliocene; P, Pleistocene. Note that *Dugong* has a complex biogeographic history and can be coded as ranging across Eastern Tethys and Indo-Australian Archipelago, see *Heritage & Seiffert (2022)*.

Tables S8–S9). We note that the paleodepth range of the Lower Al-Kharrara Member is equivalent to the present-day depths of dugong habitat along coastal Saudi Arabia, Bahrain, and Qatar. As a comparatively smaller dugongid, *Salwasiren* would have had an advantage in shallower habitats and the current data from Al Maszhabiya seem to suggest a strong taphonomic bias in this assemblage towards fossil dugongids (Figs. 2C, 2D). We think this bias is unsurprising given the abundance of modern dugong strandings adjacent to similar depositional settings today (Table S8, S9). Overall, the sedimentological and taphonomic evidence suggests that the fossil dugongids from the Al Maszhabiya bonebed are an autochthonous assemblage with little transport prior to burial.

The high abundance of fossil dugongids at Al Maszhabiya is unusual but not unique in the sirenian fossil record. Other sirenian bonebeds are known from the Eocene of
France and Spain (*Sagne, 2001*; *Díaz-Berenguer et al., 2018*), and the Middle Miocene of Mexico (*Domning, 1978*; J Velez Juarbe, pers. obs., 2025), but they have yet to be studied exhaustively (see Supplemental Information text). Other fossil marine mammal bonebeds that rival this density represent either mass strandings events (*e.g.*, the Late Miocene Cerro Ballena site in Chile, *Pyenson et al., 2014*) or hiatal surfaces with abraded and mostly isolated elements (*e.g.*, the Middle Miocene Sharktooth Hill bonebed of California, *Pyenson et al., 2009*). The monodominance of fossil dugongids has parallels to the Triassic *Shonisaurus* ichthyosaur bonebed in Nevada (*Kelley et al., 2022*), which has been interpreted as a breeding ground for giant ichthyosaurs, but there is no evidence for bimodal size distributions nor neonatal individuals at Al Maszhabiya. Even if additional dugongid species are identified, the Al Maszhabiya bonebed qualifies as a low diversity multitaxic and monodominant bonebed (*i.e.,* one taxon numerically dominant) following *Eberth, Rogers & Fiorillo's (2007)* criteria. These bonebeds are characterized by settings with low transport where aggregation behaviors are a strong influence on bonebed formation (*Brinkman, Eberth & Currie, 2007*), which is plausible given the parallels between Al Maszhabiya and modern-day dugong that aggregate socially in similar settings.

Like Al Maszhabiya, the Late Eocene Castejón de Sobrarbe (or CS-41) fossil site of northern Spain is also a monodominant sirenian bonebed, representing 6 individuals from different ontogenetic stages assigned to *Sobrarbesiren*. While the MNI and MNE values at CS-41 are comparable to Al Maszhabiya (see Table S10), the CS-41 site has been interpreted as an overbank deposit in an abandoned channel of a deltaic plain, suggesting some input from freshwater habitats. By contrast, the spatial distribution of the fossil dugongid sites in the fully marine deposit of Al Maszhabiya extends over an area 30 times greater than CS-41. If the full extent of dugongid localities from the Lower Al-Kharrara Member are included north and south, this distribution (Figs. 1C, 1D) approaches the geographic range ($\sim$200 km$^2$) of the sandstone barrier bar and shelf deposits of Wadi Hitan in Egypt (*Gingerich, 1992*; *Zalmout & Gingerich, 2012*). Notably, Al Maszhabiya exceeds the density of Late Eocene sirenian localities in Wadi Hitan by over two magnitudes (see Supplemental Information text for more details).

While the Dam Formation near the Gulf is entirely marine, outcrops of this formation further inland, especially near the type locality in eastern Saudi Arabia, represent continental depositional environments with a mix of estuarine and fluvial deposits (*Al-Saad & Ibrahim, 2002*; *Alkhaldi, Read & Al-Tawil, 2021*). Intriguingly, *Thomas et al. (1982)* reported sirenian ribs and bone fragments on the surface of two horizons from the Dam Formation near Al Sarrar (= As-Sarrar) in Saudi Arabia. Further study might reveal if these deposits can be correlated with those from marine sequences of the Dam Formation in Qatar about 200 km eastwards.

Although the spatial distribution of fossil dugongid sites at Al Maszhabiya is time-averaged over $\sim$12,500–17,000 years, the density and abundance of fossil dugongids are consistent with the live-dead data collected from surveys of living dugongs and their stranding record (Tables S8–S9). Today, dugongs aggregate in predictable locations that are conservation hotspots, such as the shallow seagrass meadows between Bahrain and Qatar. Large groups of dugongs gather and persist in this area over a decadal scale, and this

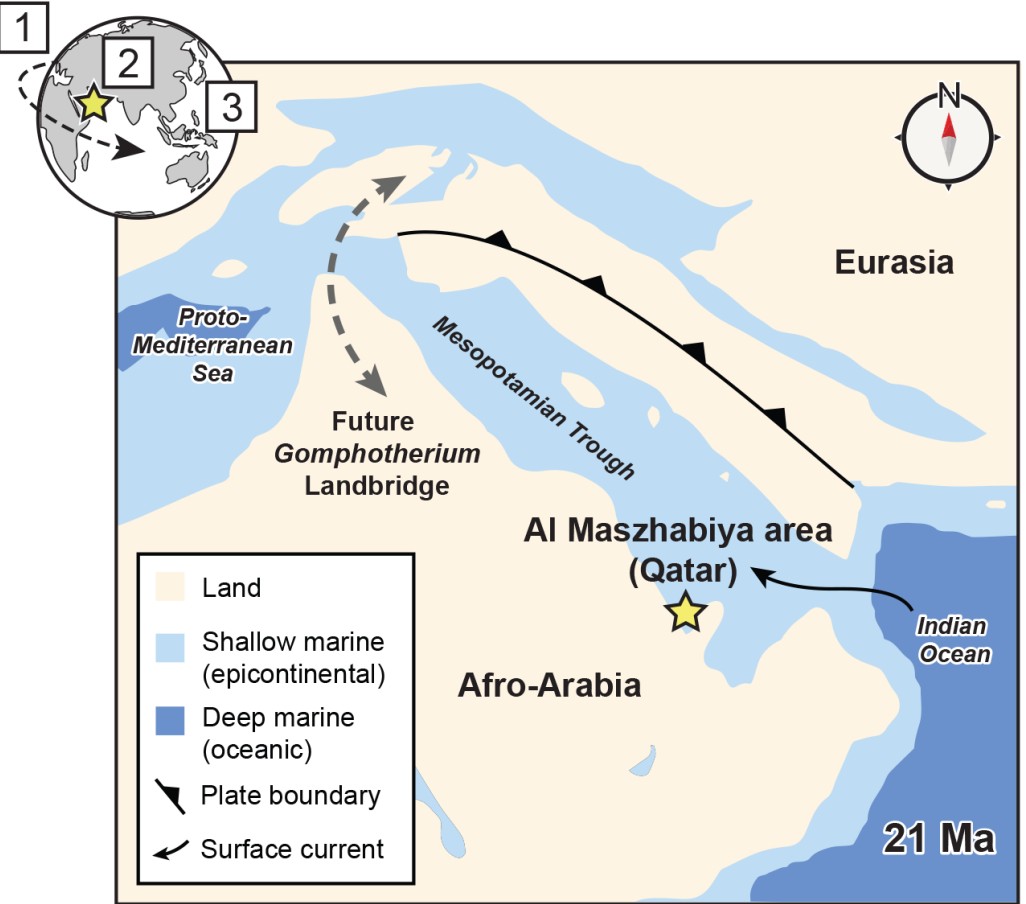

**Figure 5** **Eastern Tethys paleogeography at 21 Ma.** 1–3 on the globe shows successive jumping of biogeographic hotspots from Europe (Eocene, phase 1) to the Arabian Peninsula (Early Miocene, phase 2) to the Indo-Australian Archipelago (today, phase 3) following *Renema et al. (2008)* and others. Base map of Afro-Arabia and Eurasia modified from *Straume et al. (2025)* and *Dill & Henjes-Kunst (2007)*. The establishment of the *Gomphotherium* landbridge occurred at 19 Ma (*Rögl, 1999*; *Harzhauser et al., 2007*).

behaviorally mediated abundance is also recorded in the stranding record. We suggest that the high fidelity of the live-dead record for dugongs in an adjacent area today supports the interpretation of the Al Maszhabiya bonebed as an Early Miocene dugongid hotspot, reflecting a long-term occupation of this marginal marine habitat from the Aquitanian at least until the early Burdigalian.

## Evolution of *Salwasiren* and *Dugong*

Based on our phylogenetic analysis, *Salwasiren* belongs in a clade of dugongine dugongids that includes *Italosiren*, *Nanosiren*, *Corystosiren*, and *Dugong* (Fig. 4). *Salwasiren* is unrelated to Indian and Malagasy fossil dugongids, especially those of similar Early Miocene age in geographic proximity such as *Domningia*, *Kutchisiren*, and *Bharatisiren kachchhensis* (*Thewissen & Bajpai, 2009*). The implication of the geographic distribution of this clade is that this lineage has Tethyan origins in the early Oligocene, and then a turnover in the

Americas with *Nanosiren + Corystosiren + Dugong*, prior to the latter genus reinvading Oceania and the Gulf (Fig. 4). The Pleistocene origin of *Dugong* is likely tied to the Americas, based on the close sister relationship with *Corystosiren* and unpublished remains belonging to *Dugong* from the Pleistocene of Florida (*Domning, 2001*). The *Dugong* lineage itself has a complex biogeographic history with a subsequent dispersal to Oceania and the Indian Ocean likely *via* an Atlantic Ocean route (*Heritage & Seiffert, 2022*), given the closure of the Central American seaway by 2.8 Ma (*O'Dea et al., 2016*).

Sirenian postcranial elements are generally not sufficiently diagnostic to determine whether they belong to *Salwasiren*. The resultant profile from our rib survey at Al Maszhabiya shows a size distribution within the range of modern sirenian populations, but we cannot exclude the possibility of a multispecies assemblage, as typical of fossil dugongid communities at this time (*e.g.,* the coeval assemblage from the Khari Nadi Formation in India; *Vélez-Juarbe, Domning & Pyenson, 2012*). It is worth noting that rib comparisons are limited for sirenian multispecies assemblages because all Early Miocene fossil sirenians from India do not have associated rib material; similarly, other documented multispecies assemblages elsewhere in the world rely on cranial remains. Given that cranial remains comprise <4% of the skeletal record documented from the bonebed, we predict that discovering additional crania will be pivotal for determining whether the Al Maszhabiya bonebed represents a multispecies assemblage. Future collecting might target mandibles and tusks (*i.e.,* upper incisors), which could diagnose additional taxa as well as document more ecomorphological traits, as *Vélez-Juarbe, Domning & Pyenson (2012)* indicated.

Both the Al Maszhabiya bonebed and the broader diversification of Eastern Tethys dugongine dugongids in the Early Miocene were likely tied to high nearshore productivity with a hotspot of marine biodiversity centered over Arabia (*Renema et al., 2008*). This hotspot appears to have covered the Eastern Tethys from the Aquitanian through Burdigalian prior to the closure of the Tethys during an orogenic interval that created archipelagos north of the Arabian Plate (*Harzhauser et al., 2007*; Fig. 5). This timeframe includes most of the deposition of the Dam Formation across its extensive geographic span from Saudi Arabia to Oman (*Dill & Henjes-Kunst, 2007*). Further collecting in the Dam Formation throughout the region may yield more data about Early Miocene sirenians given the reported abundance of vertebrate fossils from these areas (*i.e., Pickford, Gommery & Al-Kindi, 2021*) and the sparsely documented sirenian material from the late Burdigalian-Langhian sequences of Saudi Arabia (*Thomas et al., 1982*).

Lastly, the Al Maszhabiya bonebed and phylogenetic position of *Salwasiren* indicate that there were multiple invasions of the nearshore ecosystems of Arabia by seagrass ecosystem engineers in the Cenozoic. Fossil marine herbivores co-occurred at Al Maszhabiya in the Early Miocene as they do today in the Gulf, but likely with different lineages; *Salwasiren* represents a distinct and unrelated stem lineage from dugongs. A Late Pleistocene dugongid occurrence from northern Qatar (*Pyenson et al., 2022*) that is morphometrically and diagnostically different from *Dugong* suggests that the Gulf was inhabited by yet another different lineage of dugongid during the Pleistocene during eustatic sea-level changes (and possibly prior to *Dugong*'s dispersal through the Indian Ocean). Reported but unpublished Eocene sirenians from Qatar represent an additional, older lineage of marine mammal

herbivore on the Arabian Peninsula, but it is unclear if these putative stem sirenians were seagrass specialists with ecomorphologies for persistent underwater grazing similar to crown lineages (*Domning & Beatty, 2007*). Thus, three separate crown dugongid lineages have inhabited nearshore environments in Qatar since the Early Miocene, likely providing an upper age bound for this lineage of ecosystem engineers. In this view, we argue that the geological persistence of carbonate platforms on the Arabian Peninsula since the Early Miocene represents an opportunity for the repeated evolution of ecological engineers in this region, initiated by a productive biological hotspot that arrived in this region around this time.

## CONCLUSIONS

The Al Maszhabiya bonebed exposed in southwestern Qatar is Aquitanian (23–21.6 Ma), representing a marine vertebrate assemblage that includes fossil sirenians, cetaceans, fishes, and sharks. The high density of fossil dugongid material from the Al Maszhabiya bonebed exceeds the abundance of fossil sirenian remains documented elsewhere in Afro-Arabia, and it likely represents one of the densest fossil sirenian sites in the world. Also, we described *Salwasiren qatarensis*, a new dugongine dugongid from the Al Maszhabiya bonebed, based mostly on an incomplete skeleton. Our phylogenetic analysis of *Salwasiren* shows that it is distantly related to today's dugongs. The distribution and taphonomy of fossil dugongids from the Al Maszhabiya bonebed occurred in similar geospatial densities with today's dugongs, suggesting that Aquitanian dugongids occupied similar ecosystem engineer roles in this region prior to the closure of the Mesopotamian Trough in the Tethys Sea at about 19 Ma. Based on these data and other fossil dugongid occurrences, we suggest that seagrass consumers have evolved repeatedly in this region (Afro-Arabia to western Asia) at least since the Early Miocene. The geologic age of the Al Maszhabiya bonebed coincides with the timing of a marine biodiversity hotspot in Arabia that has migrated from the Mediterranean region to the Indo-Australian Archipelago in the past ∼40 million years.

## NEW SPECIES REGISTRATION

The following information was supplied regarding the registration of a newly described species:

Publication LSID: urn:lsid:zoobank.org:pub:14F64E75-9D4C-4F3D-B0A6-62D4D5863B13

*Salwasiren* LSID: urn:lsid:zoobank.org:act:0D8CE42F-9CDD-452D-BD3D-CE160C0BD77F

*Salwasiren qatarensis* LSID: urn:lsid:zoobank.org:act:865814bd-0b78-403f-b64f-4f77363762a6

### Institutional Abbreviations

ARC       Archaeological Research Collections, Department of Archaeology, Qatar Museums, Doha, State of Qatar.

## ACKNOWLEDGEMENTS

At Qatar Museums, we thank H.E. Sheikha Al Mayassa bint Hamad bin Khalifa Al Thani, Mohammed Saad Al Rumaihi, and Sheikha Amna Bint Abdullaziz Al Thani for their

support of this project. We are also grateful to Sheikh Abdulazziz Al Thani and his team at the National Museum of Qatar, especially Tania Abdulmonem Al Majid and Mohammed Al Mulla for providing access and Essa Rashid Al Mansoori and Fatima Merekhan for their assistance during our field work. We also thank the Ministry of Environment and Climate Change, including the current Minister Abdullah bin Abdulaziz bin Turki Al Subaie, previous Minister Sheikh Faleh bin Nasser bin Ahmed bin Ali Al Thani, John MK Wong, Sayed J. Bukhari, and staff of the Southern Reserve for access. We thank Clare Fieseler for extensive support in the field, and we thank Azzam Al Mannai for aerial photography. Anders Pyenson, Ahmad Mujthaba Dheen Mohamed, and the staff from the US Embassy Doha provided support for fieldwork in Qatar. At NMNH, we thank Jennifer J. Hill, Scott Whittaker, and Adam Behlke for μCT scanning, Myria Perez for fossil preparation, Teresa Hsu, Darrin Lunde, and Michael McGowen for access to modern sirenian material, Scott Whittaker, Abby Telfer, Elizabeth Bruce, and Holly Little for imaging, Katherine Bemis for fossil fishes and shark assistance, and Brian Huber for microfossil assistance. We thank Erin Arndt and Chigo Ibeh at Blue Raster for their help with compiling ArcGIS data. Lastly, we are grateful to helpful comments from an anonymous reviewer and Mark T. Clementz, along with Afro-Arabian fossil vertebrate references shared by James F. Parham, which improved this manuscript.

### Funding

Fieldwork for Nicholas D. Pyenson was supported by the Kellogg Fund and the MacMillan Fund from the Department of Paleobiology. Geospatial imaging was supported by the Office of the Associate Director for Science and Chief Scientist at the Smithsonian Institution's National Museum of Natural History. Fieldwork for Christopher D. Marshall was supported by Qatar National Research Fund (NPRP No. 11S-0102-180177). The funders had no role in study design, data collection and analysis, decision to publish, or preparation of the manuscript.

### Grant Disclosures

The following grant information was disclosed by the authors:
Department of Paleobiology.
Smithsonian Institution's National Museum of Natural History.
Qatar National Research Fund: NPRP No. 11S-0102-180177.

### Competing Interests

Nicholas D. Pyenson is an Academic Editor for PeerJ.

### Author Contributions

- Nicholas D. Pyenson conceived and designed the experiments, performed the experiments, analyzed the data, prepared figures and/or tables, authored or reviewed drafts of the article, and approved the final draft.

- Ferhan Sakal conceived and designed the experiments, performed the experiments, analyzed the data, prepared figures and/or tables, authored or reviewed drafts of the article, and approved the final draft.
- Jacques LeBlanc performed the experiments, analyzed the data, authored or reviewed drafts of the article, and approved the final draft.
- Jon Blundell performed the experiments, analyzed the data, prepared figures and/or tables, authored or reviewed drafts of the article, and approved the final draft.
- Katherine D. Klim performed the experiments, analyzed the data, authored or reviewed drafts of the article, and approved the final draft.
- Christopher D. Marshall performed the experiments, analyzed the data, prepared figures and/or tables, authored or reviewed drafts of the article, and approved the final draft.
- Jorge Velez-Juarbe performed the experiments, analyzed the data, prepared figures and/or tables, authored or reviewed drafts of the article, and approved the final draft.
- Katherine Wolfe performed the experiments, analyzed the data, prepared figures and/or tables, authored or reviewed drafts of the article, and approved the final draft.
- Faisal Al-Naimi conceived and designed the experiments, performed the experiments, analyzed the data, authored or reviewed drafts of the article, and approved the final draft.

## Field Study Permissions

The following information was supplied relating to field study approvals (i.e., approving body and any reference numbers):

Fieldwork was conducted under permits from Qatar Museums (QM) and the Ministry of Environment and Climate Change (MOECC) in Qatar. The Al Maszhabiya area is registered as Heritage Area 23400 for QM and protected by the MOECC. Reports of Work describing our field seasons are archived with QM's Department of Archaeology.

## Data Availability

The data is available at Zenodo: Pyenson, N., & Sakal, F. (2025). Supporting Information for Pyenson et al. PeerJ (2025) on Al Maszhabiya bonebed from Qatar [Data set]. Zenodo. Available at https://doi.org/10.5281/zenodo.15312915.

The source datasets for 3D shapefiles and raw CT data are available at Morphosource: Available at https://www.morphosource.org/projects/000747006?locale=en.

## Supplemental Information

Supplemental information for this article can be found online at http://dx.doi.org/10.7717/peerj.20030#supplemental-information.

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
