# Peer review of "High abundance of Early Miocene sea cows from Qatar shows repeated evolution of seagrass ecosystem engineers in Eastern Tethys"

_PeerJ, doi:10.7717/peerj.20030_

## Round 0.1 · original submission · Minor Revisions

· Academic Editor

Minor Revisions

Both reviewers suggested minor changes to the paper before acceptance.

Reviewer 1 ·

Basic reporting

.

Experimental design

.

Validity of the findings

.

Additional comments

Attached

Annotated reviews are not available for download in order to protect the identity of reviewers who chose to remain anonymous.

·

Basic reporting

Overall, I found the manuscript well written, clearly organized, and highly informative. The authors also deserve credit for thoroughly documenting the permitting process for the collection and curation of these fossils—an important detail that is often overlooked. The figures are clear, well-designed, and effectively complement the text. References cited are appropriate and sufficiently cover prior research on this subject.

Editorial Suggestions
a. Lines 59–60: The current phrasing suggests dugongs occur only in the Gulf, which does not reflect their broader Indo Pacific distribution. I understand that you are referring specifically to this region, where dugong populations are restricted to the Gulf, but please clarify this in the text.
b. Permitting statement: Thank you for clearly stating the permits used for the collection and curation of these specimens.
c. Lines 124–125: Review the sentence incorporating “see Supplemental Information” for smoother integration.
d. Line 324: I was confused by the MNE value reported in the text (314) versus Fig. 2D (178). Should these values match, or do they refer to different counts (e.g., total occurrences vs. a subset from a single locality)? Clarification here would be helpful.

Experimental design

The project is largely well-designed with a clear research question and appropriate methods used. As this paper focuses on a new taphonomic study on a bonebed in Qatar that includes naming of a new species, it's clear that this represents original primary research with a well-defined and meaningful research question. No evidence of failure to meet the standards for this journal. No additional comments.

Validity of the findings

All relevant data collected as part of this project are included in the manuscript. Conlcusions drawn from these data are robust and tie back to the original research question. No additional comments.

Additional comments

The manuscript by Pyenson and co-authors documents a dense accumulation of Early Miocene–age sirenian fossils recovered from a single horizon within the Lower Al Kharrara Member of the Dam Formation in Qatar. The age, locality, and phylogenetic relationships of these specimens—tentatively regarded as a monospecific assemblage representing a new species (Salwasiren qatarensis)—make this discovery particularly noteworthy. This find helps fill a critical gap in our understanding of the transition from Paleogene to Neogene sirenian faunas and their relationship to modern taxa.

Moreover, these vertebrate remains provide an important proxy for the presence of seagrass beds, marine primary producers that are rarely preserved in the fossil record. Insights into the abundance and distribution of these ecosystems in the past can improve our understanding of how present-day seagrass communities may respond to future climate change and better constrain their role in ancient carbon cycling.

I have only two questions regarding the findings.
a. Monospecific assemblage: Given the prevalence of multiple sympatric sirenian species in nearshore assemblages throughout much of the Cenozoic (e.g., Juarbe et al., 2012), I question the emphasis placed on this being a truly monospecific bonebed. The argument based on rib thickness distributions is not entirely convincing, as ribs are among the least diagnostic elements for distinguishing sirenian taxa. Is there additional evidence that could strengthen this interpretation?

I am also curious how rib measurements from known multispecific assemblages compare to those reported here. In the supplemental information (Fig. S5), the comparison of four individuals from two living species—dugongs and manatees—shows a similar range and distribution to that of the fossil sample. While a monospecific interpretation is certainly possible, the modern data suggest the fossil assemblage could also represent at least two co-occurring species.

b. Paleoenvironmental interpretation: Could more be said about the paleoenvironment based on the dominance of sirenian remains and the apparent scarcity of other marine vertebrates? Given the inferred size of S. qatarensis, might this indicate a very shallow setting that limited the long-term presence of other marine taxa? The taphonomic bias toward sirenians is intriguing and may carry additional ecological or environmental significance worth discussing.

---

## Round 0.2 · accepted · Accept

· Academic Editor

Accept

The authors addressed all the comments to the satisfaction of the reviewer. The paper can now be accepted for publication.

·

Basic reporting

No comment.

Experimental design

No comment.

Validity of the findings

No comment.

Additional comments

The authors have thoroughly addressed the comments and questions from my initial review. I have no further comments at this time.